# A Mixed Chaotic Image Encryption Method Based on Parallel Rotation Scrambling in Rubik’s Cube Space

**DOI:** 10.3390/e27060574

**Published:** 2025-05-28

**Authors:** Lu Xu, Yun Chen, Yanlin Qin, Zhichao Yang

**Affiliations:** Department of Information Security, Naval University of Engineering, Wuhan 430033, China; feml970725@163.com (L.X.); qinyanlincool@163.com (Y.Q.); zhichao2021@126.com (Z.Y.)

**Keywords:** Rubik’s cube, chaos, Arnold transformation, image encryption

## Abstract

Most image encryption methods based on Rubik’s cube scrambling adopt the idea of cyclic shift or map the image pixels to the cube surface, not fully considering the cube’s three-dimensional (3D) properties. In response to this defect, we propose a mixed chaotic color image encryption method based on parallel rotation scrambling in 3D Rubik’s cube space. First, a seven-dimensional hyperchaotic system is introduced to generate chaotic pseudo-random integer sequences. Then, a proven lemma is applied to preprocess the red (R), green (G), and blue (B) channels of the plain image to realize the first diffusion. Next, the chaotic integer sequence is employed to control Arnold transformation, and the scrambled two-dimensional (2D) pixel matrix is converted into a 3D matrix. Then, the 3D cube is scrambled by dynamically selecting the rotating axis, layer number, and angle through the chaotic integer sequence. The scrambled 3D matrix is converted into a 2D matrix, realizing the second diffusion via exclusive OR with the chaotic matrix generated by logistic mapping. Finally, the matrices of the R, G, and B channels are combined into an encrypted image. By performing the encryption algorithm in reverse, the encrypted image can be decrypted into the plain image. A simulation analysis shows that the proposed method has a larger key space and exhibits stronger key sensitivity than some existing methods.

## 1. Introduction

With the development of network technology and artificial intelligence, the information transmission mode based on digital images has brought great convenience because of its intuitive nature, visualization features, and high information density. However, digital images involving personal privacy or secret information are faced with threats such as illegal theft and tampering in the transmission process, so it is necessary to encrypt digital images to ensure the security of information [1,2,3].

Some scholars have tried to apply traditional text information encryption methods to image encryption. However, due to the high correlation between adjacent pixels, large data capacity, and significant information redundancy, it is difficult for traditional text encryption methods to achieve efficient and secure encryption of digital images. Therefore, it is urgent to develop encryption methods suitable for digital images.

In recent years, researchers have discovered that chaos is highly effective for image encryption because of its ergodicity, aperiodicity, and pseudo-randomness. At present, chaotic image encryption has become a hotspot in the research field of image encryption [1,2,3,4,5,6,7,8,9,10,11,12,13,14,15]. To enhance encryption effects, chaotic image encryption generally includes image diffusion and image scrambling. The former approach uses chaotic pseudo-random sequences to change pixel values, while the latter leverages some scrambling methods to disrupt image pixel positions.

Most existing image diffusion methods adopt a single exclusive OR (XOR) operation in the process [13,14,15]. After rounding, a chaotic pseudo-random integer sequence is obtained, which then undergoes the XOR operation with plain-image pixel values one by one to realize image diffusion. For instance, in the Knuth–Durstenfeld shuffling algorithm based on chaotic sequences adopted in Refs. [13,14], the same random diffusion step is adopted for the red (R), green (G), and blue (B) channels of color images. Isaac et al. decomposed color images into three components (R, G, and B) and conducted discrete wavelet transformation and the same scrambling steps for each component [15]. Then, the chaotic pseudo-random sequence was used for the XOR operation with each component to achieve the purpose of diffusion.

Existing image scrambling methods mainly include Arnold transformation [16], Zigzag transform [17,18], DNA coding [19,20,21], and Rubik’s cube scrambling [22,23,24,25,26,27,28,29,30,31,32]. Among them, Rubik’s cube scrambling can realize pixel scrambling by mapping the pixels to the Rubik’s cube and utilizing its rotational property, which is characterized by simple operation and high efficiency. Some chaotic image encryption methods based on Rubik’s cube scrambling typically use the displacement characteristic of Rubik’s cubes for two-dimensional (2D) scrambling [22,23,24,25]. For example, Yang proposed an image encryption algorithm combining Rubik’s cube transformation and DNA coding by leveraging the rotation cyclic shift characteristics of the Rubik’s cube [22]. Ma adopted the cyclic shift idea of the Rubik’s cube to displace image pixels [23]. Vidhya employed the Rubik’s cube cyclic shift method for pixel scrambling [24]. Zhao et al. decomposed the six faces of a Rubik’s cube and arranged them in the 2D plane and controlled the rotation of the cube through random sequences to realize scrambling [25]. Some other chaotic image encryption methods are also based on Rubik’s cube scrambling, mapping image pixels to the cube surfaces, and then performing cube rotation [26,27,28,29,30,31]. For example, Chen et al. considered six color matrices of two color images as six faces of the Rubik’s cube, and achieved pixel scrambling by rotating the cube [26]. Zhao et al. spliced the scrambled pixel matrix and five chaotic matrices into a hexahedral structure, dynamically scrambling the Rubik’s cube through a pseudo-random sequence [27]. Wang et al. took the R, G, and B components of two images as the surfaces of the Rubik’s cube and controlled the cube’s scrambling through the pseudo-random sequence to achieve encryption [28]. Zhang mapped the image pixels to the surfaces of several third-order Rubik’s cubes, completing the spatial scrambling of image pixels by controlling the cube’s rotation [29]. Zheng et al. combined the R, G, and B components of two images into a cube and achieved image scrambling by rotating the cube [30]. Sun arranged the pixel values and keys into a hexahedron and scrambled the pixels by using a cross-axis rotation strategy [31]. However, few reports have mapped image pixels to an entire Rubik’s cube for scrambling. Notably, Gao et al. segmented multiple images by columns, superimposed multiple fixed-size planes to obtain regular cubes, and then disambiguated each tangent plane by rotating each plane counterclockwise around a fixed *Z*-axis by different angles [32].

To sum up, the existing research on chaotic image encryption based on Rubik’s cube scrambling has achieved some good results, but some shortcomings remain despite the progress:(1)In the diffusion process of existing chaotic system-based color image encryption methods, most of the R, G, and B channels in color images are diffused by the same and single XOR operation, which is similar to the steps of the grayscale image encryption method and does not make full use of the characteristics of color images, affecting the encryption effect.(2)Previous Rubik’s cube scrambling methods mostly leverage the cyclic shift feature of the Rubik’s cube or only map image pixels to the six cube surfaces. However, they do not fully consider the three-dimensional (3D) properties of the Rubik’s cube and fail to map image pixels to all the small cubes contained in the whole cube, which affects the scrambling effect. Although some studies map image pixels to the entire cube for scrambling, they simply rotate different angles around a fixed axis and fail to fully utilize the cube’s three degrees of freedom, resulting in suboptimal scrambling performance.(3)Some reports do not make full use of chaotic pseudo-random sequences generated by chaotic systems in Rubik’s cube scrambling and fail to dynamically control variables such as the cube’s rotation axis, layer, and angle, resulting in poor scrambling effects.

In response to these defects, we propose a mixed chaotic image encryption method based on parallel rotation scrambling in 3D Rubik’s cube space. First, the method generates chaotic pseudo-random integer sequences through a seven-dimensional (7D) hyperchaotic system. Then, a proven lemma is used to preprocess the R, G, and B channels of the plain image to achieve the first diffusion step. Next, the chaotic pseudo-random integer sequences are employed to control the process of Arnold transformation for scrambling, and the scrambled 2D image pixel matrix is converted into a 3D matrix. The chaotic pseudo-random integer sequences are used to dynamically select the rotation axis, rotation layer number, and rotation angle, and parallel rotation scrambling is performed in the 3D space of the Rubik’s cube. The second diffusion is realized by the XOR operation with the chaotic matrix generated by logistic mapping. Finally, the diffusion matrices of the R, G, and B channels are synthesized into an encrypted image to realize encryption. The effectiveness of the proposed mixed chaotic image encryption method is verified by comparing its performance indexes with those of existing methods through simulation experiments.

The innovation of the proposed algorithm lies in the following aspects. The pixel values of the R, G, and B channels of the color image are multiplied by different parameters, and then the same modular operation is conducted for preprocessing diffusion. Deep diffusion is carried out twice in combination with XOR diffusion, which achieves a better effect than using the same and single XOR diffusion. After mapping the image pixel values to all the small cubes contained in the Rubik’s cube, 3D space scrambling is realized. The complexity of scrambling is high, and the 3D structure of the Rubik’s cube is better utilized. The chaotic pseudo-random sequences are used to dynamically control the rotation axis, rotation layer number, and rotation angle during Rubik’s cube scrambling, realizing better randomness and more satisfactory encryption performance.

## 2. Theoretical Basis

### 2.1. Chaotic System

In this paper, we use logistic mapping and a 7D hyperchaotic system to generate chaotic pseudo-random sequences.

The expression of logistic mapping is as follows:(1)$$x_{n+1}=\mu x_{n}\left(1-x_{n}\right),$$
where $x_{n}$ denotes the state variable of the system and μ represents the control parameter. When $\mu \in \left[0,4\right]$ and $x_{n}\in \left(0,1\right)$, logistic mapping appears chaotic [14].

The 7D hyperchaotic system is expressed as follows [33]:(2)$$\left\{\begin{array}{l} {\dot{y}}_{1}=a\left(y_{2}-y_{1}\right)+y_{4}+y_{7},\\ {\dot{y}}_{2}=-cy_{2}-y_{1}y_{3}+y_{6},\\ {\dot{y}}_{3}=-b+y_{1}y_{2},\\ {\dot{y}}_{4}=-y_{2}-y_{5},\\ {\dot{y}}_{5}=dy_{2}+y_{4},\\ {\dot{y}}_{6}=-ey_{1}+y_{2},\\ {\dot{y}}_{7}=-y_{2}+y_{6}, \end{array}\right.$$
where $y_{1},y_{2},y_{3},y_{4},y_{5},y_{6},y_{7}$ are the state variables of the system, and a,b,c,d,e represent the control parameters. When $\left(a,b,c,d,e\right)=\left(10,100,2.7,2,3\right)$ and the initial variable values are set to $\left(y_{1}\left(0\right),\right.y_{2}\left(0\right),y_{3}\left(0\right),y_{4}\left(0\right),\left.y_{5}\left(0\right),y_{6}\left(0\right),y_{7}\left(0\right)\right)=\left(1,1,1,1,1,1,1\right)$, the system exhibits a hyperchaotic state [33].

**Remark** **1.**
*Logistic mapping is hereinafter used to generate 2D chaotic pseudo-random sequence matrices for diffusion, and the 7D hyperchaotic system is employed to create chaotic pseudo-random sequences for preprocessing parameters, Arnold transformation times, and Rubik’s cube scrambling control parameters in the encryption algorithm.*


### 2.2. Arnold Transformation

Arnold transformation [32] is a method proposed by V. J. Arnold in the study of ergodic theory. In Arnold transformation, if the pixel coordinates of an N×N image are $x,y\in \left\{0,1,2,3,\ldots ,N-1\right\}$, discrete Arnold transformation can be employed as follows:(3)$$\left[\begin{array}{l} x^{\prime }\\ y^{\prime } \end{array}\right]=\left[\begin{array}{cc} 1 & 1\\ 1 & 2 \end{array}\right]\left[\begin{array}{l} x\\ y \end{array}\right]\mathrm{mod}\;N,$$
where x and y are the pixel coordinates in the plain image, $x^{\prime }$ and $y^{\prime }$ denote the pixel coordinates in the new image, and “mod” represents the function of taking remainders.

Arnold inverse transformation is described by(4)$$\left[\begin{array}{l} x\\ y \end{array}\right]={\left[\begin{array}{cc} 1 & 1\\ 1 & 2 \end{array}\right]}^{-1}\left[\begin{array}{l} x^{\prime }\\ y^{\prime } \end{array}\right]\mathrm{mod}\;N.$$

Arnold transformation features periodicity and can obtain the plain image after certain iterations. Through the code given in Ref. [34], the 2D Arnold transformation period of images under a varying order N can be obtained, as shown in Table 1. It can be observed that the 2D Arnold transformation period is related but not proportional to the image size.

As can be seen from Table 1, the 2D Arnold transformation period T is 383 in this algorithm since the 512 × 512 × 3 Lena standard color image is used as the plain image in the simulation.

### 2.3. Fundamentals of Remainders

**Lemma** **1.***For* $\left(a\times b\right)\mathrm{mod}256=c$*,* $a,b,c\in \left[0,255\right]$*,* a,b,c∈Z*, and* $\left(a,256\right)=1$*, if* a *and* c *are known, then* b *has a unique solution.*

**Proof of Lemma** **1.**If $\left(a\times b\right)\mathrm{mod}256=c$, $a,b,c\in \left[0,255\right]$, a,b,c∈Z, and $\left(a,256\right)=1$, then $a^{-1}$ exists that satisfies $a\cdot a^{-1}\equiv 1\mathrm{mod}256$. Therefore, the original formula is equivalent to $a^{-1}\cdot a\cdot b\equiv a^{-1}\cdot c\mathrm{mod}256$, that is, $b\equiv a^{-1}\cdot c\mathrm{mod}256$. Since $a^{-1}$ is unique, b is unique. QED. □

**Remark** **2.**
*In encryption and decryption methods, Lemma 1 can be used in image preprocessing and inverse processing to ensure the uniqueness of decomposition and synthesis of the R, G, and B channels.*


According to the Euler function, 128 integers in $\left[0,255\right]$, namely, $\left\{1,3,5,7,9,11,13,\right.15,$$\left.\;\ldots ,251,253,255\right\}$, are coprime with 256. However, if the first element 1 is substituted into Equation (7) below, the preprocessed pixel value may not change, which may ultimately lead to subpar image encryption. Therefore, we remove the first element 1 in the plain set and obtain the 127-element key set of $K_{3}=\left\{3,5,7,9,11,13,15,\ldots ,251,253,255\right\}$ needed by the proposed encryption algorithm.

## 3. Mixed Chaotic Image Encryption Method Based on Parallel Rotation Scrambling in 3D Rubik’s Cube Space

For any color image with a size of M×N, we design the mixed chaotic image encryption method based on parallel rotation scrambling in 3D Rubik’s cube space as follows.

**Step 1**: Let $l=\max\left(M,N\right)$. Select the key $K_{1}=\left(\mu ,x_{0},i_{1},i_{q}\right)=\left(3.998,\right.0.720,$${\left(ceil\left(\sqrt{{\left(ceil\left(\sqrt[{3}]{l^{2}}\right)\right)}^{3}}\right)\right)}^{2}+\left.1000,1000\right)$, which specifies the parameter, initial value, iteration time, and number of elimination groups of logistic mapping given in Equation (1), and then iterate logistic mapping several times. A chaotic pseudo-random sequence with a size of ${\left(ceil\left(\sqrt{{\left(ceil\left(\sqrt[{3}]{l^{2}}\right)\right)}^{3}}\right)\right)}^{2}+1000$ is generated, and the first 1000 values are removed to prevent the transient effect, obtaining the chaotic pseudo-random sequence x. To convert this into a pixel integer value of $\left[0,255\right]$, the following processing is employed:(5)$$X=floor\left(x\times 10^{16}\right)\mathrm{mod}256,$$
where $ceil\left(\cdot \right)$ is an upward rounding function and $floor\left(\cdot \right)$ represents a downward rounding function.

Then, the above chaotic pseudo-random integer sequence X is transformed into a chaotic matrix $A_{6}$ with a size of $ceil\left(\sqrt{{\left(ceil\left(\sqrt[{3}]{l^{2}}\right)\right)}^{3}}\right)\times ceil\left(\sqrt{{\left(ceil\left(\sqrt[{3}]{l^{2}}\right)\right)}^{3}}\right)$.

**Step 2**: Select the key $K_{2}=\left(a,b,c,d\right.,e,y_{1}\left(0\right),y_{2}\left(0\right),y_{3}\left(0\right),y_{4}\left(0\right),y_{5}\left(0\right),y_{6}\left(0\right),y_{7}\left(0\right),$$\left.i_{2},i_{q},i^{\prime }\right)\left.=\left(10,100,2.7,\right.2,3,1,1,1,1,1,1,1,{\left(ceil\left(\sqrt[{3}]{l^{2}}\right)\right)}^{3}+1000,1000,2\right)$, which specifies the parameters, initial values, iteration times, number of elimination groups, Arnold transformation parameter, and preprocessing parameter of the 7D hyperchaotic system depicted in Equation (2). A chaotic pseudo-random sequence with a size of ${\left(ceil\left(\sqrt[{3}]{l^{2}}\right)\right)}^{3}+1000$ is generated, and the first 1000 groups are similarly removed to prevent the transient effect, obtaining 7 pseudo-random sequences: $y_{1}\left(i\right),y_{2}\left(i\right),y_{3}\left(i\right),y_{4}\left(i\right),y_{5}\left(i\right),y_{6}\left(i\right),y_{7}\left(i\right)$.

**Step 3**: Integerize the chaotic pseudo-computer sequences $y_{1}\left(i\right),y_{2}\left(i\right),y_{3}\left(i\right),y_{4}\left(i\right),$$y_{5}\left(i\right),y_{6}\left(i\right),y_{7}\left(i\right)$ with Equation (6), below, to attain the integer sequences $Y_{1}\left(i\right),Y_{2}\left(i\right),Y_{3}\left(i\right),$$Y_{4}\left(i\right),Y_{5}\left(i\right),Y_{6}\left(i\right),Y_{7}\left(i\right)$:(6)$$\left\{\begin{array}{l} Y_{1}\left(i\right)=floor\left(y_{1}\left(i\right)\times 10^{16}\right)\mathrm{mod}127+1,\\ Y_{2}\left(i\right)=floor\left(y_{2}\left(i\right)\times 10^{16}\right)\mathrm{mod}127+1,\\ Y_{3}\left(i\right)=floor\left(y_{3}\left(i\right)\times 10^{16}\right)\mathrm{mod}127+1,\\ Y_{4}\left(i\right)=floor\left(y_{4}\left(i\right)\times 10^{16}\right)\mathrm{mod}\left(T-1\right)+1,\\ Y_{5}\left(i\right)=floor\left(y_{5}\left(i\right)\times 10^{16}\right)\mathrm{mod}3,\\ Y_{6}\left(i\right)=floor\left(y_{6}\left(i\right)\times 10^{16}\right)\mathrm{mod}ceil\left(\sqrt[{3}]{l^{2}}\right)+1,\\ Y_{7}\left(i\right)=floor\left(y_{7}\left(i\right)\times 10^{16}\right)\mathrm{mod}3, \end{array}\right.$$
where $i=1,2,3,\ldots ,{\left(ceil\left(\sqrt[{3}]{l^{2}}\right)\right)}^{3}$, and T denotes the 2D Arnold transformation period corresponding to images of different sizes (Table 1).

**Step 4**: Read the color plain image P of size M×N, and divide it into three channels: $R_{P},G_{P},B_{P}$.

**Step 5**: Taking the $R_{P}$ channel image as an example, zero-fill it to obtain the $R_{1}$ channel image with a size of l×l, $K_{4}=\left(M,N\right)$. In this way, $G_{1}$ and $B_{1}$ channel images with a size of l×l can also be obtained.

**Step 6**: Substitute the value of $i^{\prime }$ into the chaotic pseudo-random integer sequences $Y_{1}\left(i\right),Y_{2}\left(i\right),Y_{3}\left(i\right)$ to obtain $Y_{1}\left(i^{\prime }\right),Y_{2}\left(i^{\prime }\right),Y_{3}\left(i^{\prime }\right)$. The $Y_{1}\left(i^{\prime }\right)$th, $Y_{2}\left(i^{\prime }\right)$th, and $Y_{3}\left(i^{\prime }\right)$th values of $K_{3}=\left\{3,5,7,9,11,13,\right.$$\left.15,\ldots ,251,253,255\right\}$ are then selected as the preprocessing parameters of the pixel values of the $R_{1}$, $G_{1}$, and $B_{1}$ channels, respectively. The pixel values of the $R_{2}$, $G_{2}$, and $B_{2}$ channels are obtained by preprocessing each channel using the following equations:(7)$$\left\{\begin{array}{l} R_{2}=\left(K_{3}\left(Y_{1}\left(i^{\prime }\right)\right)\times R_{1}\right)\mathrm{mod}256,\\ G_{2}=\left(K_{3}\left(Y_{2}\left(i^{\prime }\right)\right)\times G_{1}\right)\mathrm{mod}256,\\ B_{2}=\left(K_{3}\left(Y_{3}\left(i^{\prime }\right)\right)\times B_{1}\right)\mathrm{mod}256. \end{array}\right.$$

**Step 7**: Convert the $R_{2}$ channel image into a pixel matrix $A_{1}$, and substitute the value of $i^{\prime }$ into the sequence $Y_{4}\left(i\right)$ to attain $Y_{4}\left(i^{\prime }\right)$. Then, take $Y_{4}\left(i^{\prime }\right)$ as the number of the Arnold transformation and perform Arnold transformation on $A_{1}$ to obtain the transformed matrix $A_{2}$.

**Step 8**: Read $A_{2}$ as array a in row order and fill it with ${\left(ceil\left(\sqrt[{3}]{l^{2}}\right)\right)}^{3}-l^{2}$ zeroes to obtain array b. Then, read b as $ceil\left(\sqrt[{3}]{l^{2}}\right)$ 2D matrices with a size of $ceil\left(\sqrt[{3}]{l^{2}}\right)\times ceil\left(\sqrt[{3}]{l^{2}}\right)$. These 2D matrices are stacked sequentially from top to bottom to form a 3D pixel matrix $A_{3}$ with a size of $ceil\left(\sqrt[{3}]{l^{2}}\right)\times ceil\left(\sqrt[{3}]{l^{2}}\right)\times ceil\left(\sqrt[{3}]{l^{2}}\right)$.

**Step 9**: Perform Rubik’s cube scrambling on $A_{3}$ in the following manner: $Y_{5}\left(i\right)$ determines whether the rotation axis of $A_{3}$ is axis X($Y_{5}\left(i\right)=0$), axis Y($Y_{5}\left(i\right)=1$), or axis Z($Y_{5}\left(i\right)=2$); $Y_{6}\left(i\right)$ controls the layer that $A_{3}$ rotates to be the $Y_{6}\left(i\right)$th layer; $Y_{7}\left(i\right)$ decides whether the rotation angle of the selected layer is 90° ($Y_{7}\left(i\right)=0$), 180° ($Y_{7}\left(i\right)=1$), or 270° ($Y_{7}\left(i\right)=2$). The rotation direction is clockwise, and the operation starts from i=1 to $i={\left(ceil\left(\sqrt[{3}]{l^{2}}\right)\right)}^{3}$. Taking the fifth-order Rubik’s cube as an example, the specific method is shown in Figure 1.

**Step 10**: Read the scrambled 3D pixel matrix $A_{4}$ as array c in reverse order according to the method in step 8, and add ${\left(ceil\left(\sqrt{{\left(ceil\left(\sqrt[{3}]{l^{2}}\right)\right)}^{3}}\right)\right)}^{2}-{\left(ceil\left(\sqrt[{3}]{l^{2}}\right)\right)}^{3}$ zeros after c to obtain array d. Read d in reverse order as a 2D pixel matrix $A_{5}$, and perform XOR with $A_{6}$ to generate the final encrypted image $R_{C}$ of the R channel.

Similarly, the encryption operations from steps 7 to 10 are performed on the $G_{2}$ and $B_{2}$ channels, obtaining the final encrypted images $G_{C}$ and $B_{C}$ of the G and B channels.

**Step 11**: The encrypted images, $R_{C}$, $G_{C}$, and $B_{C}$, of the three channels are merged and converted into an encrypted color image C.

The overall process is illustrated in Figure 2. The pseudocode is as in Algorithm 1.
**Algorithm 1.** Mixed chaotic image encryption method based on parallel rotation scrambling in 3D Rubik’s cube space.**Input:** Image P of size M×N and keys $K_{1},K_{2},K_{3},K_{4}$
**Output:** Image C
1: Generate chaotic sequence x using Logistic mapping;2: Convert chaotic sequence x to integer sequence X after removing the first 1000 values of x;3: Transform X into chaos matrix A6;4: sequenceLength = K1(3);5: Logisticmap = zeros(sequenceLength, 1);6: Logisticmap(1) = K1(2);7: for i = 2:sequenceLength8:    logisticmap(i) = K1(1) × logisticmap(i − 1) × (1 − logisticmap(i − 1));9: end10: Logisticmap = Logisticmap(1001:end);11: X = mod(floor(Logisticmap × 1 × 10^16^), 256);12: A6 = reshape(X, sqrt(length(X)), sqrt(length(X)));13: Generate hyperchaotic sequence y using the 7D hyperchaotic system;14: Convert hyperchaotic sequence y to integer sequence Y after removing the first 1000 values of y;15: dy = zeros(7, 1);16: y(1) = K26:12;17: seqLength = K213;18: for i = 2:seqLength19:    dy1 = K21× (y2 − y1) + y4 + y7;20:    dy2 = −K23×y2 − y1×y3 + y6;21:    dy3 = −K22 + y1×y2;22:    dy4 = −y2 − y5;23:    dy5 = K24×y2 + y4;24:    dy6 = −K25×y1 + y2;25:    dy7 = −y2 + y6;26: end27: y = y(1001:end, :);28: Y1 = mod(floor(abs(y(:, 1)) × 1× 10^16^), 127) + 1; 29: Y2 = mod(floor(abs(y(:, 2)) × 1× 10^16^), 127) + 1; 30: Y3 = mod(floor(abs(y(:, 3)) × 1× 10^16^), 127) + 1; 31: Y4 = mod(floor(abs(y(:, 4)) × 1× 10^16^), 382) + 1; 32: Y5 = mod(floor(abs(y(:, 5)) × 1× 10^16^), 3); 33: Y6 = mod(floor(abs(y(:, 6)) × 1× 10^16^), ceil(l^2/3^) + 1; 34: Y7 = mod(floor(abs(y(:, 7)) × 1× 10^16^), 3); 35: Preprocess the plain image; 36: R1 = double(P(:, :, 1));37: G1 = double(P(:, :, 2));38: B1 = double(P(:, :, 3));39: for i = 1:numel(R1)40:    R2(i: numel(R1)) = mod(K3(Y1(2)) × R1(i: numel(R1)), 256);41:    G2(i: numel(R1)) = mod(K3(Y2(2)) × G1(i: numel(R1)), 256);42:    B2(i: numel(R1)) = mod(K3(Y3(2)) × B1(i: numel(R1)), 256);43: end44: Perform Arnold transformation;45: [rows, cols] = size(R2);46: N = rows; 47: R3 = R2;48: G3 = G2;49: B3 = B2;50: for iter = 1:Y4K21551:    for x = 0:rows-152:        for y = 0:cols-153:            x_new = mod((x + y), N);54:            y_new = mod((x + 2*y), N);55:            R3(x_new + 1, y_new + 1) = R2(x + 1, y + 1);56:            G3(x_new + 1, y_new + 1) = G2(x + 1, y + 1);57:            B3(x_new + 1, y_new + 1) = B2(x + 1, y + 1);58:        end59:    end60:    R2 = R3;61:    G2 = G3;62:    B2 = B3;63: end64: Convert transformed matrix A2 to 3D pixel matrix A3;65: Scramble the Rubik’s cube A3;66: R4 = zeros(cube_size, cube_size, cube_size);67: G4 = zeros(cube_size, cube_size, cube_size);68: B4 = zeros(cube_size, cube_size, cube_size);69: for i = 1:cube_size70:    for j = 1:cube_size71:        for k = 1:cube_size72:            index = (i − 1)×cube_size^2^ + (j − 1)×cube_size + k;73:            if index ≤ numel(R3)74:                R4(i,j,k) = R3(index);75:                G4(i,j,k) = G3(index);76:                B4(i,j,k) = B3(index);77:            end78:        end79:    end80: end81: for i = 1:ceil((length(Y5)) 82:    axis = Y5(i) + 1; 83:    layer = Y6(i); 84:    rotation = Y7(i) + 1;85:    R4 = rubiks_cube_rotate(R4, axis, rotation, layer);86:    G4 = rubiks_cube_rotate(G4, axis, rotation, layer);87:    B4 = rubiks_cube_rotate(B4, axis, rotation, layer);88: end89: Convert 3D matrix A4 into 2D matrix A5 after XOR with chaotic matrix A6;90: Merge encrypted channels into color encrypted image C.

## 4. Mixed Chaotic Image Decryption Method Based on Parallel Rotation Scrambling in 3D Rubik’s Cube Space

The mixed chaotic image decryption method based on parallel rotation scrambling in 3D Rubik’s cube space is the reverse process of the above encryption process; it is performed in the following steps.

**Step 1**: Let $l=\max\left(M,N\right)$. Iterate the logistic mapping described in Equation (1) by using the values in $K_{1}=\left(\mu ,x_{0},i_{1},i_{q}\right)\left.=\left(3.998,0.720,\right.{\left(ceil\left(\sqrt{{\left(ceil\left(\sqrt[{3}]{l^{2}}\right)\right)}^{3}}\right)\right)}^{2}+1000,1000\right)$ as the parameter, initial value, iteration time, and number of elimination groups of logistic mapping, respectively. Then, obtain the chaotic pseudo-random sequence x after removing the first 1000 values. Process x according to Equation (5) to turn it into the interval of $\left[0,255\right]$, and then transform it into a chaotic matrix $A_{7}$ with a size of $ceil\left(\sqrt{{\left(ceil\left(\sqrt[{3}]{l^{2}}\right)\right)}^{3}}\right)\times ceil\left(\sqrt{{\left(ceil\left(\sqrt[{3}]{l^{2}}\right)\right)}^{3}}\right)$.

**Step 2**: Iterate the 7D hyperchaotic system in Equation (2) by using the values in $K_{2}=\left(a,\right.$$b,c,d,e,y_{1}\left(0\right),\left.y_{2}\left(0\right),y_{3}\left(0\right),y_{4}\left(0\right),y_{5}\left(0\right),y_{6}\left(0\right),y_{7}\left(0\right),i_{2},i_{q},i^{\prime }\right)=\left(10,100,2.7,2,\right.3,1,1,1,1,1,1,1,$${\left(ceil\left(\sqrt[{3}]{l^{2}}\right)\right)}^{3}+1000,\left.1000,2\right)$ as the parameters, initial values, iteration times, number of elimination groups, Arnold transformation parameter, and preprocessing parameter of the 7D hyperchaotic system, respectively. Then, remove the first 1000 groups of values to obtain 7 chaotic pseudo-random sequences $y_{1}\left(i\right),y_{2}\left(i\right),y_{3}\left(i\right),y_{4}\left(i\right),y_{5}\left(i\right),y_{6}\left(i\right),y_{7}\left(i\right)$.

**Step 3**: Integerize the chaotic pseudo-random sequences $y_{1}\left(i\right),y_{2}\left(i\right),y_{3}\left(i\right),y_{4}\left(i\right),$$y_{5}\left(i\right),y_{6}\left(i\right),y_{7}\left(i\right)$ via Equation (6) to obtain the chaotic pseudo-random integer sequences $Y_{1}\left(i\right),Y_{2}\left(i\right),Y_{3}\left(i\right),Y_{4}\left(i\right),Y_{5}\left(i\right),Y_{6}\left(i\right),Y_{7}\left(i\right)$. Substitute the value of $i^{\prime }$ into the sequences $Y_{1}\left(i\right),Y_{2}\left(i\right),Y_{3}\left(i\right),Y_{4}\left(i\right)$ to obtain $Y_{1}\left(i^{\prime }\right),Y_{2}\left(i^{\prime }\right),$$Y_{3}\left(i^{\prime }\right),Y_{4}\left(i^{\prime }\right)$.

**Step 4**: Divide the encrypted color image into three channels, $R_{C}$, $G_{C}$, and $B_{C}$, and decrypt the image of each channel using steps 5 to 10 to obtain the initial decrypted image of each channel.

**Step 5**: Take the $R_{C}$ channel image as an example, convert it into a pixel matrix $A_{C}$, and then perform an XOR operation between it and $A_{7}$ to generate a 2D pixel matrix $A_{8}$.

**Step 6**: Read $A_{8}$ as array e in row order, remove the first ${\left(ceil\left(\sqrt{{\left(ceil\left(\sqrt[{3}]{l^{2}}\right)\right)}^{3}}\right)\right)}^{2}-$${\left(ceil\left(\sqrt[{3}]{l^{2}}\right)\right)}^{3}$ zeros, and then read it as array f in reverse order. Read f sequentially as $ceil\left(\sqrt[{3}]{l^{2}}\right)$ 2D matrices with a size of $ceil\left(\sqrt[{3}]{l^{2}}\right)\times ceil\left(\sqrt[{3}]{l^{2}}\right)$, and stack these 2D matrices from top to bottom to form a 3D pixel matrix $A_{9}$ with a size of $ceil\left(\sqrt[{3}]{l^{2}}\right)\times$$ceil\left(\sqrt[{3}]{l^{2}}\right)\times ceil\left(\sqrt[{3}]{l^{2}}\right)$.

**Step 7**: Perform a reverse-order 3D Rubik’s cube disarrangement of $A_{9}$ in the following way: $Y_{5}\left(i\right)$ decides whether the rotation axis of $A_{9}$ is axis X($Y_{5}\left(i\right)=0$), axis Y($Y_{5}\left(i\right)=1$), or axis Z($Y_{5}\left(i\right)=2$); $Y_{6}\left(i\right)$ determines the layer $A_{9}$ rotates to be the $Y_{6}\left(i\right)$th layer; $Y_{7}\left(i\right)$ controls whether the rotation angle of the selected layer is 90° ($Y_{7}\left(i\right)=0$), 180° ($Y_{7}\left(i\right)=1$), or 270° ($Y_{7}\left(i\right)=2$). The rotation direction is counterclockwise, and the operation is from i=1 to $i={\left(ceil\left(\sqrt[{3}]{l^{2}}\right)\right)}^{3}$.

**Step 8**: Read the scrambled 3D pixel matrix $A_{10}$ into array g in reverse order according to the method in step 6. Remove the last ${\left(ceil\left(\sqrt[{3}]{l^{2}}\right)\right)}^{3}-l^{2}$ zeros after g to obtain array h, and read h as a 2D pixel matrix $A_{11}$ with a size of l×l.

**Step 9**: Taking $Y_{4}\left(i^{\prime }\right)$ as the number of times to perform Arnold inverse transformation, perform Arnold inverse transformation on $A_{11}$ and obtain the transformed pixel matrix $A_{12}$. Then, $A_{12}$ is transformed into the $R_{12}$ channel image.

**Step 10**: In $K_{3}=\left\{3,5,7,9,11,13,15,\ldots ,251,253,255\right\}$, the $Y_{1}\left(i^{\prime }\right)$th, $Y_{2}\left(i^{\prime }\right)$th, and $Y_{3}\left(i^{\prime }\right)$th values are selected as the inverse processing parameters of the pixel values of the three channels $R_{12}$, $G_{12}$, and $B_{12}$, respectively. The image of each channel is inversely processed, and the pixel values of the images in the $R_{P^{\prime }}$, $G_{P^{\prime }}$, and $B_{P^{\prime }}$ channels are obtained as follows:(8)$$\left\{\begin{array}{l} R_{P^{\prime }}=\left({K_{3}}^{-1}\left(Y_{1}\left(i^{\prime }\right)\right)\times R_{12}\right)\mathrm{mod}256,\\ G_{P^{\prime }}=\left({K_{3}}^{-1}\left(Y_{2}\left(i^{\prime }\right)\right)\times G_{12}\right)\mathrm{mod}256,\\ B_{P^{\prime }}=\left({K_{3}}^{-1}\left(Y_{3}\left(i^{\prime }\right)\right)\times B_{12}\right)\mathrm{mod}256. \end{array}\right.$$

**Step 11**: The initial decrypted images of the three channels are merged and converted into a color image $P^{\prime }$ with a size of l×l. According to $K_{4}=\left(M,N\right)$, the last $l^{2}-M\times N$ pixels with a value of 0 are removed to obtain the final decrypted image P, with a size of M×N.

The overall process is illustrated in Figure 3.

## 5. Simulation Results and Analysis

In this paper, MATLAB R2022a is used to simulate and verify the proposed mixed chaotic image encryption method. The operating environment is as follows: 13th Gen Intel(R) Core(TM) i7-13700H@2.40 GHz, 32 GB memory, a 64-bit operating system, and an X64-based processor. To evaluate the encryption effect of this method intuitively, the Lena standard color image with a size of 512 × 512 × 3 is selected as the experimental encryption and decryption object. As can be seen from Figure 4, the encrypted image exhibits random noise, and the decrypted image is consistent with the plain image, demonstrating that the proposed method can effectively realize image encryption and decryption.

### 5.1. Key Sensitivity Analysis

Key sensitivity is an important index to measure the security of encryption algorithms. To judge the sensitivity of the proposed algorithm to the change of key, we decrypt the encrypted image with the correct key and the wrong key after slightly changing the correct key. (The wrong key is $K_{1}^{\prime }=\left(\mu ,x_{0},i_{1},i_{q}\right)=\left(3.998,\right.0.720+10^{-15},$${\left(ceil\left(\sqrt{{\left(ceil\left(\sqrt[{3}]{l^{2}}\right)\right)}^{3}}\right)\right)}^{2}+\left.1000,1000\right)$ while $K_{2},K_{3},K_{4}$ remains unchanged.) The results are shown in Figure 5. It can be seen that even though the difference between the two keys is slight, the correct image cannot be decrypted. Therefore, the proposed algorithm has good key sensitivity.

### 5.2. Key Space Analysis

The key space of encryption algorithms must be at least greater than 2^100^ to effectively resist violent attacks. The keys used in this algorithm are as follows: $K_{1}=\left(\mu ,x_{0},i_{1},i_{q}\right)=$$\left(3.998,\right.0.720,{\left(ceil\left(\sqrt{{\left(ceil\left(\sqrt[{3}]{l^{2}}\right)\right)}^{3}}\right)\right)}^{2}+\left.1000,1000\right),K_{2}=\left(a,b,c,d\right.,e,y_{1}\left(0\right),y_{2}\left(0\right),y_{3}\left(0\right),y_{4}\left(0\right),$$y_{5}\left(0\right),y_{6}\left(0\right),y_{7}\left(0\right),\left.i_{2},i_{q},i^{\prime }\right)=\left(10,100,2.7,2,3,1,1,1,1,1,1,1\right.,\left.{\left(ceil\left(\sqrt[{3}]{l^{2}}\right)\right)}^{3}+1000,1000,2\right),K_{3}=\left\{3,\right.$$5,7,9,11,13,15,\ldots ,251,253\left.255\right\},K_{4}=\left(M,N\right)$. Using a computer with a 64-bit operating system, the floating-point precision can reach 10^−16^, so the sensitivities of logistic mapping and the 7D hyperchaotic system to parameters and initial values are 10^−16^. The parameters of the 7D hyperchaotic system are fixed, so the integerization of pseudo-random sequences is also accurate to 16 decimal places. It can be estimated that the key space size of this algorithm is at least ${\left(10^{16}\right)}^{3}\times {\left(10^{16}\right)}^{13}\times 127^{3}\approx 2^{871}$, which is far greater than 2^100^. This confirms the algorithm’s strong ability to resist violent attacks.

Table 2 compares the key space size of different algorithms, revealing that the proposed algorithm has a larger key space and better image encryption effect.

### 5.3. Histogram Analysis

As shown in Figure 6, the plain image exhibits significant non-uniform distribution characteristics in the histogram, and its pixel values are concentrated in specific intervals, reflecting a statistical law. The histogram of the image encrypted by the proposed method shows an approximately uniform distribution, which indicates that the encryption process eliminates the statistical characteristics of the plain image through sufficient scrambling and diffusion. Therefore, attackers cannot use the histogram distribution law to infer the original information, which verifies that the proposed encryption algorithm can resist statistical analysis attacks.

### 5.4. Correlation Analysis of Adjacent Pixels

As can be seen from Figure 7, the adjacent pixels of the plain image show strong correlations, while the adjacent pixels of the encrypted image have few correlations. To further evaluate the correlation of adjacent image pixels, the correlation coefficients of adjacent pixels of the plain and encrypted images in horizontal, vertical, and diagonal directions are calculated. The correlation coefficients of the proposed algorithm are compared with those of other algorithms, as shown in Table 3.

Table 3 reveals that the correlation coefficients of adjacent pixels of the image encrypted by the proposed method are far lower than those of the plain image and generally smaller than those of other methods. These results demonstrate the superior performance of our image encryption method.

### 5.5. Information Entropy Analysis

Table 4 compares the information entropy of the Lena color plain image, the image encrypted by the proposed encryption method, and the images obtained from other comparative methods. Compared with the images encrypted by other methods, the image encrypted by our method exhibits information entropy closer to the ideal value of 8, indicative of its superior encryption performance.

### 5.6. Robustness Analysis

In practical applications, image encryption methods should have the ability to resist shearing attacks and noise attacks. To evaluate the robustness of the proposed algorithm, 1/16, 1/8, and 1/4 of the encrypted image area are cut before decryption. Gaussian noise with a variance of 0.001, 0.005, and 0.01 is added to the encrypted image and then the images are decrypted. The results are shown in Figure 8 and Figure 9, respectively. It can be observed that the decrypted images are affected to varying degrees after cropping or adding noise, but some information of the plain image can be recovered when the cut image part is small or the noise pollution intensity is low. This shows that the proposed algorithm has a strong anti-attack ability and can maintain basic recoverability in harsh channel environments.

### 5.7. Time Comparison

To verify the influence of the Rubik’s cube scrambling times on the final encryption effect of the proposed method, we vary the scrambling time of the Rubik’s cube and employ performance indexes such as the encryption and decryption time, correlation coefficient of adjacent pixels, and information entropy for encryption effect analysis according to the actual size of the Lena color image. In the encryption algorithm, the Rubik’s cube scrambling times are $i={\left(ceil\left(\sqrt[{3}]{l^{2}}\right)\right)}^{3}$, and the actual value of i is 262,144 after substituting the Lena color image parameters. Then, the Rubik’s cube scrambling times are set to $\frac{i}{10}$, $\frac{i}{100}$, and $\frac{i}{1000}$ (the actual values are 26,215, 2622, and 263, respectively) to encrypt the same Lena color image. The obtained performance indexes are compared in Table 5.

As can be seen from Table 5, the correlation coefficient and information entropy of adjacent pixels of the final encrypted image vary slightly under different Rubik’s cube scrambling times, while the encryption and decryption times increase significantly with an increase in scrambling times. This means that when the difference in encryption performance is negligible, the encryption algorithm with a lower scrambling time will have a shorter encryption time and higher encryption efficiency.

To eliminate the interference of other encryption steps in the proposed method and further study the influence of the Rubik’s cube scrambling time on image encryption effects, the same Lena color image is scrambled with different Rubik’s cube scrambling times, and image encryption performance metrics such as the scrambling time and correlation coefficient of adjacent pixels are used for evaluation.

As can be seen from Table 6, for scrambling times of i or $\frac{i}{10}$, the correlation coefficient of adjacent pixels of the scrambled image is low, and there is little difference between the two scenarios. However, the former case has a much shorter scrambling time than the latter. Therefore, if the difference in encryption performance is negligible, the encryption algorithm with a lower scrambling time should be selected, which can yield a shorter encryption time and higher encryption efficiency.

## 6. Conclusions

Existing chaotic image encryption methods based on Rubik’s cube scrambling suffer from single diffusion, insufficient use of the 3D properties of Rubik’s cubes, and deficient adoption of chaotic pseudo-random sequences generated by chaotic systems. In response to these disadvantages, this study proposes a mixed chaotic image encryption method based on parallel rotation scrambling in 3D Rubik’s cube space. The innovations of this method are as follows: First, the pixel values of the R, G, and B channels of a color image are multiplied by different parameters, and then the same modular operation is performed for preprocessing diffusion. Combined with XOR diffusion, deep diffusion is performed twice. Second, the pixel values of the image are mapped to the small cubes contained in the whole Rubik’s cube, and then the 3D space is scrambled, which exhibits high complexity and makes better use of the characteristics of the cube’s 3D structure. Third, chaotic pseudo-random sequences are used to dynamically control the rotation axis, layer, and angle of the Rubik’s cube, which achieves better randomness and more satisfactory encryption effects. Simulation experiments compare the performance indexes and prove the effectiveness of this mixed chaotic image encryption method compared to some existing methods.

## Figures and Tables

**Figure 1 entropy-27-00574-f001:**
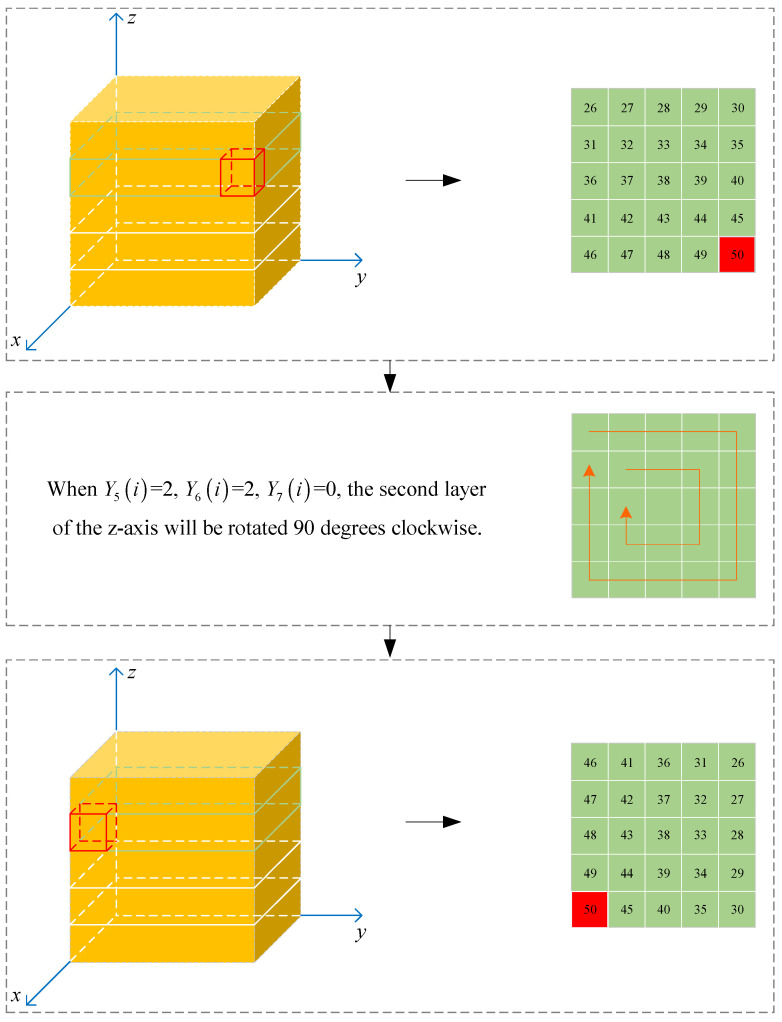
Rubik’s cube scrambling.

**Figure 2 entropy-27-00574-f002:**
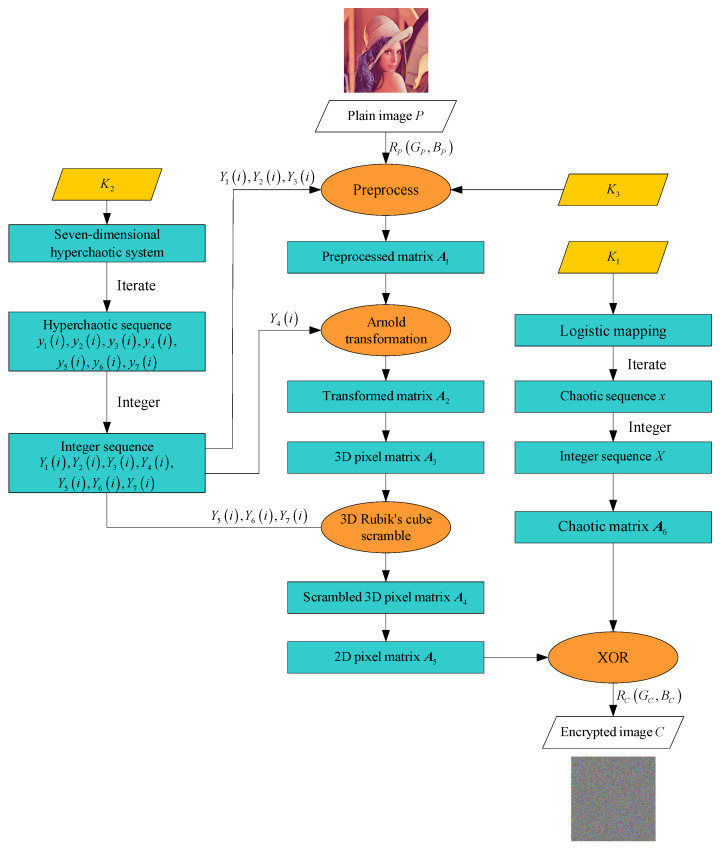
Encryption flowchart.

**Figure 3 entropy-27-00574-f003:**
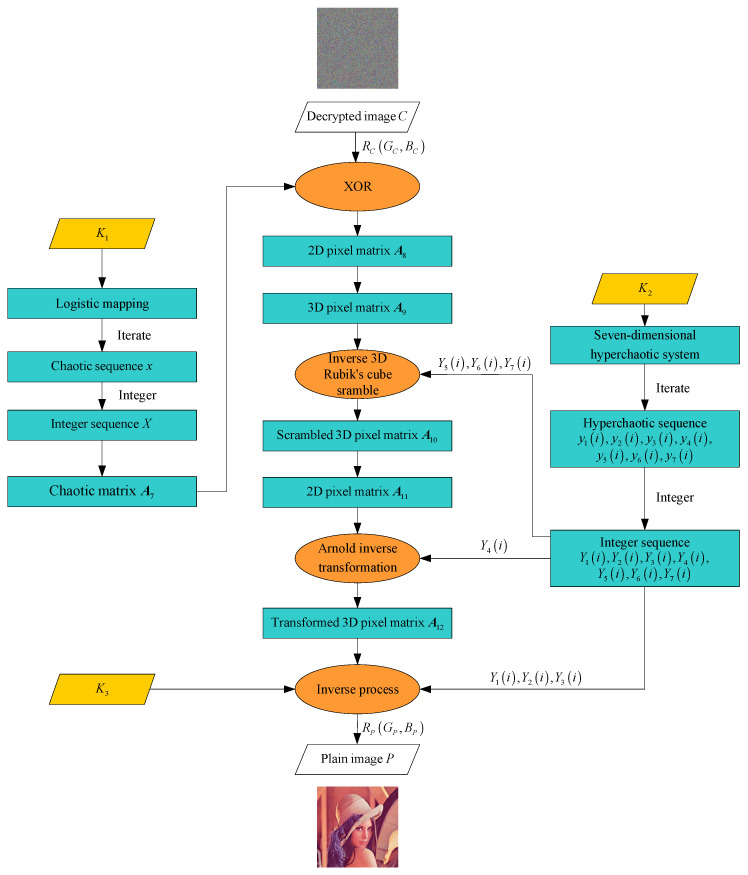
Decryption flowchart.

**Figure 4 entropy-27-00574-f004:**
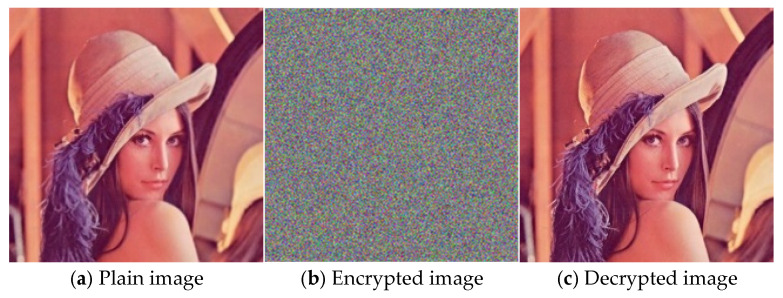
Plain image, encrypted image, and decrypted image.

**Figure 5 entropy-27-00574-f005:**
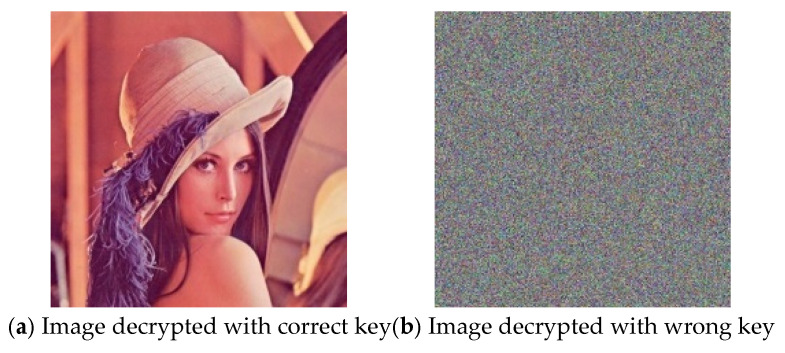
Key sensitivity test results.

**Figure 6 entropy-27-00574-f006:**
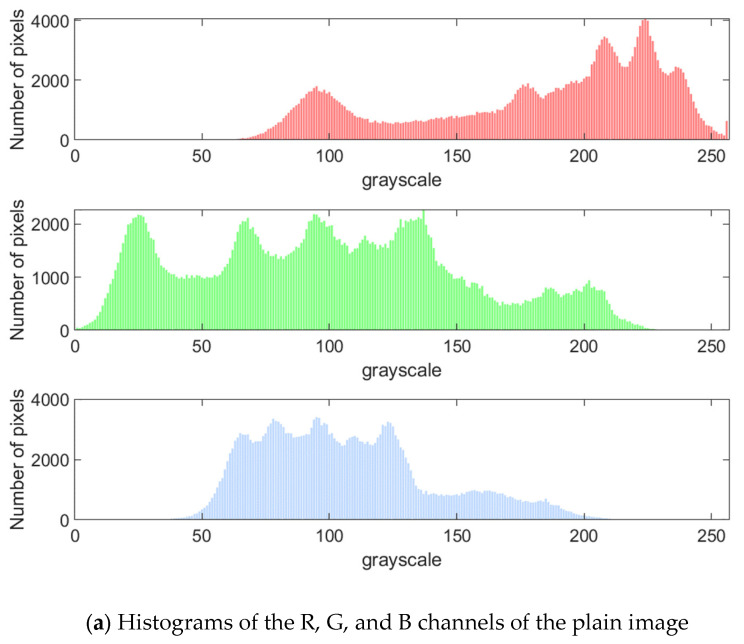
Histograms of the plain image and encrypted image.

**Figure 7 entropy-27-00574-f007:**
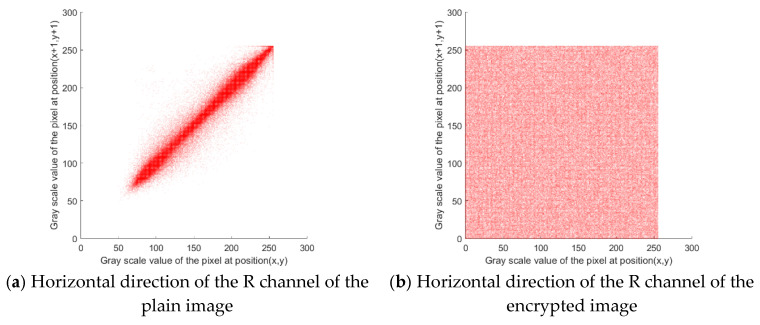
Comparison of the adjacent pixel correlations between the plain and encrypted images.

**Figure 8 entropy-27-00574-f008:**
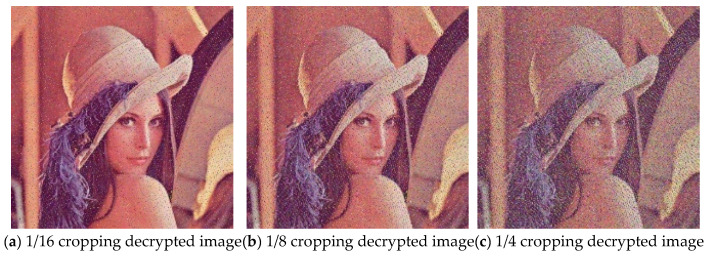
Decrypted images subjected to cropping attacks with different proportions.

**Figure 9 entropy-27-00574-f009:**
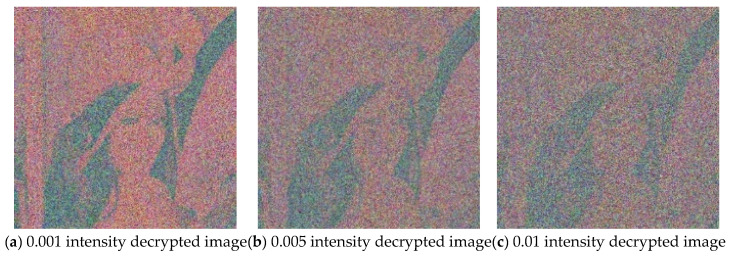
Decrypted images after adding Gaussian noise with different variances.

**Table 1 entropy-27-00574-t001:** The 2D Arnold transformation periods corresponding to images under different orders [34].

N	2	4	8	10	25	50	100	125	256	512
T	3	3	6	30	50	150	150	250	192	383

**Table 2 entropy-27-00574-t002:** Comparison of key spaces for different algorithms.

Method	Number of Keys
Ours	2^871^
Ref. [31]	2^199^
Ref. [25]	2^212^
Ref. [27]	2^450^
Ref. [14]	2^512^

**Table 3 entropy-27-00574-t003:** Comparison of correlation coefficients between adjacent pixels.

Method	Horizontal	Vertical	Diagonal
Plain image	0.9818	0.9681	0.9517
Ours	−0.001600	−0.000939	0.000243
Ref. [14]	0.003496	−0.000815	−0.001055
Ref. [27]	−0.006267	−0.001400	0.001067
Ref. [29]	0.001269	−0.004160	0.001162
Ref. [31]	−0.001100	−0.002733	−0.001000

**Table 4 entropy-27-00574-t004:** Comparison of information entropy.

Method	R	G	B	Average
Plain image	7.2682	7.5901	6.9951	7.28447
Ours	7.9992	7.9994	7.9994	7.99933
Ref. [31]	7.9956	7.9949	7.9953	7.99527
Ref. [27]	7.9976	7.9973	7.9974	7.99743
Ref. [14]	7.9993	7.9994	7.9991	7.99927
Ref. [29]	7.9993	7.9993	7.9992	7.99927

**Table 5 entropy-27-00574-t005:** Comparison of encryption and decryption times and encryption effects for different cube scrambling times.

Number of Disruptions		i	$\frac{i}{10}$	$\frac{i}{100}$	$\frac{i}{1000}$
Encryption time		1138.45 s	113.50 s	15.21 s	5.82 s
Decryption time		1067.61 s	133.79 s	24.72 s	14.95 s
Neighboring pixel correlation coefficient	Horizontal	−0.00160	0.00131	0.00088	0.00105
	Vertical	−0.00094	−0.00286	0.00022	0.00111
	Diagonal	0.00024	0.00125	−0.00151	−0.00006
Information entropy	R	7.9992	7.9993	7.9993	7.9993
	G	7.9994	7.9993	7.9993	7.9993
	B	7.9994	7.9992	7.9992	7.9993
	Average	7.99933	7.99927	7.99927	7.99930

**Table 6 entropy-27-00574-t006:** Comparison of scrambling time and encryption effects for different cube scrambling times.

Number of Disruptions		i	$\frac{i}{10}$	$\frac{i}{100}$	$\frac{i}{1000}$
Scrambling time		2199.74 s	226.41 s	11.30 s	1.95 s
Neighboring pixel correlation coefficient	Horizontal	0.03680	0.03527	0.05531	0.23495
	Vertical	0.02790	0.03696	0.06371	0.27084
	Diagonal	0.02549	0.02210	0.02946	0.09948

## Data Availability

The data used to support the findings of this study are included in the article.

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
