# Peer review of "A Mixed Chaotic Image Encryption Method Based on Parallel Rotation Scrambling in Rubik’s Cube Space"

_entropy, 2025, doi:10.3390/e27060574_

Round 1
Reviewer 1 Report
Comments and Suggestions for Authors
In this paper the authors are interested in an efficient method for image encryption-decryption. They propose a hybrid method: chaotic image-color encryption method based on parallel rotation decryption in Rubik's cube space. The results shown by the authors are interesting and the comparison they show with other methods evidences the effectiveness of the proposed method, however, in my opinion, the article presents different drawbacks in the writing and presentation of the results.
In the abstract, the authors describe the encryption process without mentioning the results or the best result obtained, moreover, in the article they present the image decryption method, which is not mentioned in the abstract.
In the introduction, they lack references to the relevance of image encryption in general (line 31).
In lines 54-93, they describe relevant and previous work on image decryption methods, this paragraph is very long and the wording could be improved.
The 7-dimensional hyperchaotic system (equation 2) is not clear whether the authors propose it or it is all from reference [33], which I did not find for the review, check the reference.
In the rationale section there is a lack of proper wording.
In equation (6) the punctuation is inadequate.
The method is described in detail between lines 193-251, however, after line 254 the keys K_i are mentioned as inputs, which are given above but are not clear in an equation (summarized).
In Figure 1, the colors used could be changed and improve the visualization.
In Figure 6, the labels should not be spliced with the results shown in the graphs.
In Section 5.3 the wording could be improved.
In general, the wording of the article can be improved, the proper formatting is not respected in the document structure (spaces between paragraphs, sections, subsections, etc.), in addition, the mathematical mode inserted in the text is not properly formatted.
I didn't find the reference [34]
Finally, it is suggested to revise the words
Pseudorandom-->pseudo-random
Matrixes-->matrices
Because of its content, I recommend that the article be accepted with major corrections.
Author Response
Comments 1: In the abstract, the authors describe the encryption process without mentioning the results or the best result obtained, moreover, in the article they present the image decryption method, which is not mentioned in the abstract.
Response 1: Thank you for your advice. We have made the following improvements to the abstract: 1) Supplementary results: improved results of performance indicators. 2) Add the decryption process: By performing the encryption algorithm in reverse, the encrypted image can be decrypted into a plain image. 3) Streamlining the encryption process: Streamlining the details of the encryption process, highlighting the innovations and achievements. Please refer to page 1, paragraph 1, lines 8-24 of the revised manuscript for details.
Comments 2: In the introduction, they lack references to the relevance of image encryption in general (line 31).
Response 2: Thank you for your advice. We have supplemented the references there. Please refer to page 1, paragraph 2, line 33 of the revised manuscript for details.
Comments 3: In lines 54-93, they describe relevant and previous work on image decryption methods, this paragraph is very long and the wording could be improved.
Response 3: Thank you for your valuable comments on the expression of the manuscript. We have optimized the relevant paragraphs by streamlining the content and optimizing the language expression. Please refer to page 2, paragraph 3, lines 56-85 of the revised manuscript for details.
Comments 4: The 7-dimensional hyperchaotic system (equation 2) is not clear whether the authors propose it or it is all from reference [33], which I did not find for the review, check the reference.
Response 4: The equation 2 is all from reference [33], and I have indicated it in the revised manuscript. Please refer to page 3, paragraph 8, line 136 of the revised manuscript for details. At the same time, I will upload reference [33] with the attachment.
Comments 5: In the rationale section there is a lack of proper wording.
Response 5: The wording has been revised for better clarity. Please refer to the red part on page 2, line 86 to page 5, line 177 of the revised manuscript for details.
Comments 6: In equation (6) the punctuation is inadequate.
Response 6: Thank you for your important advice. We have checked and modified equation (6) and other equations. Please refer to equation (2), (6), (7) and (8) of the revised manuscript for details.
Comments 7: The method is described in detail between lines 193-251, however, after line 254 the keys K_i are mentioned as inputs, which are given above but are not clear in an equation (summarized).
Response 7: Thank you for your attention to the rigor of algorithm description. I have explained the keys involved after line 254 in detail to ensure clarity. Please refer to page 10, paragraph 2, lines 253-256 and page 11, paragraph 1, lines 260-264 of the revised manuscript for details.
Comments 8: In Figure 1, the colors used could be changed and improve the visualization.
Response 8: I have changed the colors of Figure 1 and improved the visualization.
Comments 9: In Figure 6, the labels should not be spliced with the results shown in the graphs.
Response 9: I have removed the labels from Figure 6.
Comments 10: In Section 5.3 the wording could be improved.
Response 10: I have improved the wording of Section 5.3. Please refer to page 14, paragraph 3, lines 340-347 of the revised manuscript for details.
Comments 11: In general, the wording of the article can be improved, the proper formatting is not respected in the document structure (spaces between paragraphs, sections, subsections, etc.), in addition, the mathematical mode inserted in the text is not properly formatted.
Response 11: We have carefully reviewed and revised the wording, document structure, and mathematical mode to ensure accuracy and clarity. All text modifications have been tracked using Microsoft Word’s review mode.
Comments 12: I didn't find the reference [34]
Response 12: Reference [34] is a Chinese reference. I will upload it with the attachment.
Comments 13: it is suggested to revise the words
Pseudorandom-->pseudo-random
Matrixes-->matrices
Response 13: Thank you for pointing out the language problems in the manuscript in detail. We have checked the full text sentence by sentence, corrected grammar, spelling and punctuation errors, and checked the citation format of the references.
Reviewer 2 Report
Comments and Suggestions for Authors
A paper entitled "A mixed chaotic image encryption method based on parallel rotation scrambling in Rubik’s cube space."
The abstract provides appreciable problem statement and methodology but lacks compelling research findings.
The literature review of the existing works covers the key areas of focus of the paper.
Correct all the typos and/or grammatical and spelling errors e.g., "matrixes" is "matrices."
What are the constraints affecting the dynamic selection of the rotation axis, rotation layer number, and rotation angle by the chaotic pseudo-random integer sequences? What is the latency of the parallel rotation scrambling within the 3D space of the Rubik’s cube?
The authors should provide a table of comparison of the proposed technique versus the conventional implementations
Comments on the Quality of English LanguageAppreciable English language but the typos and grammatical errors should be corrected.
Author Response
Comments 1: The abstract provides appreciable problem statement and methodology but lacks compelling research findings.
Response 1: Thank you for your valuable advice. We have supplemented the abstract. After stating the existing problems in this field and the methods adopted in this manuscript, the results of key performance indicators are briefly expounded. Please refer to page 1, paragraph 1, lines 8-24 of the revised manuscript for details.
Comments 2: The literature review of the existing works covers the key areas of focus of the paper.
Response 2: Although the existing works have achieved some good research results, there are still the following shortcomings: 1) In the diffusion process of existing encryption methods, R, G and B channels of color images are diffused by the same and single XOR operation respectively. 2) The existing scrambling methods of the Rubik's cube mostly use the cyclic shifting characteristics of the Rubik's cube or just map the image pixels to the surfaces of the six faces of the Rubik's cube to scramble, without fully considering the three-dimensional properties of the Rubik's cube. 3) Some achievements did not make full use of the chaotic pseudo-random sequence to participate in the Rubik's cube scrambling, and did not dynamically control the variables such as the rotation axis, the number of rotation layers and the rotation angle of the Rubik's cube. In view of the above defects, the innovation of the method proposed in this manuscript lies in: Firstly, we multiple the pixel values of R, G and B channels of color images with different parameters respectively. Then performing the same modular operation for preprocessing diffusion, and combining XOR diffusion, performing two deep diffusion. Secondly, the image pixel values are mapped to all the small cubes contained in the whole Rubik's cube, and then the three-dimensional space is scrambled, which makes better use of the characteristics of the three-dimensional cube structure of the Rubik's cube. Thirdly, using chaotic pseudo-random sequence to dynamically control the rotation axis, number of rotation layers and rotation angle of Rubik's cube scrambling has better randomness and encryption effect.
Comments 3: Correct all the typos and/or grammatical and spelling errors e.g., "matrixes" is "matrices."
Response 3: Thank you for pointing out the language problems in the manuscript in detail. We have checked the full text sentence by sentence, corrected grammar, spelling and punctuation errors, and checked the citation format of the references.
Comments 4: What are the constraints affecting the dynamic selection of the rotation axis, rotation layer number, and rotation angle by the chaotic pseudo-random integer sequences?
Response 4: Thank you for your valuable advice. Aiming at this problem, we explain as follows: By iterating over the seven-dimensional hyperchaotic system, the pseudo-random sequence y5, y6, and y7 is generated. And then they are integer-processed to get Y5, Y6, and Y7 respectively. The value of Y5 is 0, 1, and 2. The value of Y6 is 1, 2, 3, …, ceil(l^(2/3)). The value of Y7 is 0, 1, and 2. In the process of Rubik's cube rotation, it can be controlled by three independent parameters, namely, the rotation axis, the number of rotation layers and the rotation angle. The rotation axis includes X axis, Y axis and Z axis. The number of rotation layers is the side length of the Rubik's cube, including 1, 2, 3, …, ceil(l^(2/3)). And the rotation angles include 90°, 180° and 270°. (Rotation angles of 0° and 360° has no scrambling effect, so 0° and 360° are not considered here.) The specific selection of Rubik's cube rotation parameters is as follows: determines whether the rotation axis is axis X (Y5=0), axis Y (Y5=1), or axis Z (Y5=2); Y6 controls the number of rotation layers, and the number of rotation layers is the value of Y6; Y7 decides whether the rotation angle of the selected layer is 90°(Y7=0), 180°(Y7=1), or 270°(Y7=2).
Comments 5: What is the latency of the parallel rotation scrambling within the 3D space of the Rubik’s cube?
Response 5: Thank you for your valuable advice. The latency of the parallel rotation scrambling within the 3D space of the Rubik’s cube you mentioned, in our opinion, is the time to scramble the three-dimensional Rubik's cube. It specifically refers to the time needed from the beginning to the end of the Rubik's cube scrambling. Table 6 of the manuscript summarizes the scrambling time of different scrambling times.
Comments 6: The authors should provide a table of comparison of the proposed technique versus the conventional implementations
Response 6: Thank you for your valuable advice. In Section 5, the experimental results and analysis of the proposed technique are compared with those of the conventional implementations mentioned in the summary in terms of key space, correlation coefficient of adjacent pixels and information entropy. The specific results are shown in Table 2, 3 and 4.
Comments 7: The English could be improved to more clearly express the research.
Response 7: We sincerely appreciate your valuable feedback on the language of our manuscript. To enhance the quality of our English writing, we have implemented the following improvements: 1) Professional Language Editing – The manuscript has been meticulously proofread and revised by English language experts. 2) Terminology Standardization – Technical terms have been carefully reviewed to ensure accuracy and consistency across the text. 3) Sentence Structure Improvement – Overly complex or lengthy sentences have been streamlined for greater clarity and readability.
Round 2
Reviewer 2 Report
Comments and Suggestions for Authors
The revision appreciably addresses most of my comments.
Comments on the Quality of English LanguageGood.